# The Spectrum of Disease-Associated Alleles in Countries with a Predominantly Slavic Population

**DOI:** 10.3390/ijms25179335

**Published:** 2024-08-28

**Authors:** Grigoriy A. Yanus, Evgeny N. Suspitsin, Evgeny N. Imyanitov

**Affiliations:** 1Laboratory of Molecular Diagnostics, St. Petersburg State Pediatric Medical University, 194100 St. Petersburg, Russia; octavedoctor@yandex.ru; 2Department of Medical Genetics, St. Petersburg State Pediatric Medical University, 194100 St. Petersburg, Russia; evgeny.suspitsin@gmail.com; 3Department of Tumor Growth Biology, N.N. Petrov Institute of Oncology, 197758 St. Petersburg, Russia

**Keywords:** medical genetics, founder effect, founder mutation, inherited disease, germline mutation, Slavs, genetic burden, next-generation sequencing, pathogenic variants

## Abstract

There are more than 260 million people of Slavic descent worldwide, who reside mainly in Eastern Europe but also represent a noticeable share of the population in the USA and Canada. Slavic populations, particularly Eastern Slavs and some Western Slavs, demonstrate a surprisingly high degree of genetic homogeneity, and, consequently, remarkable contribution of recurrent alleles associated with hereditary diseases. Along with pan-European pathogenic variants with clearly elevated occurrence in Slavic people (e.g., *ATP7B* c.3207C>A and *PAH* c.1222C>T), there are at least 52 pan-Slavic germ-line mutations (e.g., *NBN* c.657_661del and *BRCA1* c.5266dupC) as well as several disease-predisposing alleles characteristic of the particular Slavic communities (e.g., Polish *SDHD* c.33C>A and Russian *ARSB* c.1562G>A variants). From a clinical standpoint, Slavs have some features of a huge founder population, thus providing a unique opportunity for efficient genetic studies.

## 1. Introduction

Founder populations are an invaluable resource for medical genetic studies [1]. If a given hereditary disorder is present in a relatively isolated community, the spectrum of disease-associated alleles is usually limited to one or a few pathogenic variants, which are characterized by elevated population frequency. Consequently, founder communities provide advantageous conditions for the identification of disease-predisposing genes, the analysis of gene-specific penetrance, the recruitment of allele carriers, the organization of screening, and many other research and healthcare activities. Some founder populations are represented by relatively small groups of people who managed to preserve their cultural and religious identity while residing on well-defined territories within large countries. For example, Mennonite and Amish communities, while originating from a small number of founders of Western European descent, have accumulated a high number of recurrent pathogenic alleles due to a very high rate of endogamy [2]. There are examples when founder communities have emerged due to a ban on interethnic marriage. For instance, Ashkenazi Jews, who initially settled in Europe and then spread to North America, are characterized by the persistence of a high number of unique genetic diseases and recurrent pathogenic variants [3]. In addition, inhabitants of some countries, like Iceland or Finland, experienced substantial geographic isolation in the past due to natural barriers and/or tough climate conditions resulting in the uniqueness of their genetic background [4,5]. All the examples mentioned above deal with communities with a moderate number of inhabitants ranging from a few thousand to a few million people. Some larger populations, e.g., Saudi Arabia, have their “genetic load” unevenly distributed due to the high rate of consanguineous marriages and accumulation of particular autosomal-recessive diseases within certain tribes or large families [6].

Countries with predominantly Slavic populations were parts of large multiethnic empires and experienced significant historical turbulences in the past. Despite these circumstances, Slavic people appear to have an unexpectedly large repertoire of recurrent pathogenic variants, thus suggesting some level of genetic isolation. The cataloging of genetic diseases in Slavic countries has so far been performed with a lesser comprehension as compared to Western Europe and North America. Here, we review the data on the persistence of founder pathogenic alleles in Slavic populations.

## 2. Language, Brief History, Demography, and Genetics of Slavic Populations

The first mention of the Slavs appeared in Byzantine chronicles and is dated by the VI century CE [7]. Linguistic studies suggest that Balto-Slavic folks originated from other Indo-Europeans approximately 7000–4500 years ago. Later, around 3500–2500 years ago, Baltic and Slavic language groups separated from each other [8]. 

The division for Eastern Slavs, Western Slavs (Poles, Czechs, Slovaks, Sorbs, Kashubians), and Southern Slavs (Serbs, Croats, Slovenians, Bosnians, Macedonians, Bulgarians) occurred around the V–VI centuries CE [9,10]. It was not until the IX–X centuries when the centralized states, inhabited mostly by Slavs, like Great Moravia, Poland, and Kievan Rus’, were mentioned in chronicles. During the Middle Ages, the Eastern Slavs divided into Russians, Ukrainians, and Belarusians. Archeological and linguistic studies allow tracking the dispersal of early Slavs [11] (Figure 1).

Currently, there are 13 Eastern European countries with a predominance of Slavic-speaking population: Belarus, Bosnia, Bulgaria, Croatia, Czech Republic, Montenegro, North Macedonia, Poland, Russia, Slovakia, Slovenia, Serbia, and Ukraine.

Genetic studies of the countries with predominantly Slavic-speaking populations show that the majority of the Western and Eastern Slavs are relatively genetically homogenous, while Southern Slavs demonstrate greater diversity [12]. Poles have very significant similarities with Eastern Slavs. Czechs and, to a lesser extent, Slovaks demonstrate substantial genetic similarity to Germans and other Central Europeans, while Southern Slavs are relatively close to both Northern Slavs and non-Slavic-speaking Balkan populations [13].

The degree of ethnic homogeneity in a number of Eastern European countries remains high. For example, according to the 2021 census, more than 95% of the population of Poland is represented by ethnic Poles by self-determination (Table 1); it needs to be acknowledged that the ethnic diversity of Poland was much more pronounced before World War II [14]. The modern population of Russia includes 180 non-Slavic ethnic groups, many of which are quite numerous (e.g., Tatars, Yakuts, Chechens, etc.); however, these communities are residing in particular regions and largely preserve their ethnic identity. According to the 2010 census, Slavic people comprise 79.5% of the total population of the Russian Federation (Table 1). Many countries (e.g., Germany, Austria, Hungary) share borders with Slavic states and therefore have a substantial proportion of people with Slavic roots (Table 1).

There are also regions of relatively compact residence of Slavic ethnic isolates, e.g., Sorbs (Lusatian Serbs) living in Saxony and Brandenburg [13]. Most of the Slavic countries experienced several waves of mass emigration to Western Europe and North America, particularly in the past century. Consequently, the share of people of Slavic origin is approximately 6% in the USA and 11% in Canada (Table 1). Thus, there are millions of Slavs living outside their “native” communities, and they carry the set of clinically relevant disease-associated alleles common in their homelands [15,16,17,18]. Several predominantly Slavic countries, for example, Poland, Ukraine, and Belarus, used to have large Ashkenazi Jewish settlements. Consequently, Eastern European Jews have a substantial admixture of Slavic alleles. In addition, there are several isolated religious communities currently situated in the USA and Canada, who initially migrated from German-speaking countries to Slavic states to avoid prosecution, and who also adopted a significant number of Slavic alleles.

**Table 1 ijms-25-09335-t001:** The number of Slavs by self-determination according to the Census data.

Country	Number of Slavic Inhabitants	Total	% of Slavs	Year of Census
Countries with a predominantly Slavic population				
Russia [19]	113,545,845	142,856,536	79.5%	2010
Ukraine [20]	46,500,300	48,254,113	96.4%	2001
Poland [21]	37,643,682	38,036,118	99%	2021
Belarus [22]	9,145,526	9,413,446	97.2%	2019
Czech Republic [23]	6,618,842	10,524,167	62.9% ^3^	2021
Serbia [24]	5,717,514	6,647,003	86%	2022
Bulgaria [25]	5,163,637	6,519,789	79.2%	2021
Slovakia [26]	4,641,011	5,449,270	85.2%	2021
Croatia [27]	3,731,006	3,871,833	96.4%	2021
Bosnia and Herzegovina [28]	3,401,105	3,531,159	96.3%	2016
Slovenia [29]	1,742,243	1,964,036	88.7%	2002
North Macedonia [30]	1,179,952	2,097,319	56.3%	2021
Montenegro [31]	470,798	620,029	75.9%	2011
Selected non-Slavic countries				
USA ^1^ [32]	16,529,466	282,200,000	5.9%	2000
Canada ^1^ [33]	3,843,590	34,460,065	11.2%	2019
Kazakhstan [34]	3,860,055	18,631,779	20.7%	2020
Germany [35]	2,787,860	83,200,000	3.4%	2020
UK ^2^ [36]	853,000	64,596,800	1.3%	2014
Uzbekistan [37]	820,700	32,120,500	2.6%	2017
Romania [38]	125,165	20,121,641	0.6%	2013
Total	263,209,488			

^1^ In these countries’ census, a person can claim two ancestries; therefore, the sum of all reported entities by ancestry is greater than the total population number. ^2^ The detailed information on reported ancestry is not publicly available, but Polish people (n = 853,000) are said to be the most common non-British nationality in UK. ^3^ If we would remove more than 3.3 million inhabitants, who have not stated their ethnicity, we would observe 6,618,842 Slavs out of 7,203,109 (91.8%) population.

## 3. Methodology of the Search and the Use of Terminology

Despite the controversial genetic ancestry of some Slavic-speaking populations, throughout this review, we use the terms “Slavic-speaking” and “Slavic” population interchangeably, and we have included in our review all populations considered Slavic due to linguistic and/or historical reasons. Furthermore, we use the terms “founder” and “recurrent” alleles interchangeably, although a high frequency of a given variant may be observed both due to the reproductive success of a common ancestor and because of the independent emergence of the same hotspot mutation in multiple individuals. The formal proof of the founder nature of the variant requires haplotyping, which has not yet been performed for all variants mentioned in this article. Furthermore, we use the terms “mutation” and “pathogenic variant” interchangeably across the text, although the latter definition is currently considered more preferable for the description of hereditary disorders.

Relevant articles, which were published before January 5, 2024, and described ethnicity-specific mutational landscape of inherited conditions, were searched for in the PubMed database using the following queries:

((nationality/isolate* OR country/region**) AND (genetic disease OR hereditary syndrome OR hereditary disease OR deficiency OR disorder) AND (recurrent variant OR recurrent mutation OR founder variant OR founder mutation OR founder effect)) OR ((nationality/isolate* OR country/region**) AND (“germline mutation” OR “germline variant” OR “hereditary disease” OR “genetic syndrome” OR “hereditary syndrome” OR “genetic disease” OR “recurrent variant” OR “recurrent mutation” OR “gene defects” OR “genetic lesions” OR “founder variant” OR “founder mutation” OR “founder effect”)) OR (“nationality/isolate* founder mutation” OR “nationality/isolate* founder allele” or “nationality/isolate* founder variant”). 

Where nationality/isolate* is: Russian; Polish; Belarusian, Belorusian, Belorussian or Belarus or Byelorussian; Ukrainian; Czech; Slovak or Slovakian; Slovene or Slovenian; Serbian or Serb; Croat or Croatian; Bosnian or Bosniak; Bulgarian; Macedonian; Montenegrin; Sorbian or Lusatian or Kashubian or Kashub or Rusyn or Ruthenian or Karpatian or Carpathian; Balt or Baltic or Lithuanian or Latvian or Estonian and country/region** is: Russia or Russian Federation; Poland or Polish Republic; Belarus or Belorussia or Byelorussia; Ukraine; Czech Republic; Slovakia; Slovenia; Serbia; Croatia; Bosnia; Bulgaria; Macedonia or North Macedonia; Montenegro; Kashubia or Carpathia or Karpatia; Baltic States or Lithuania or Latvia or Estonia or Hungary

The papers were manually curated to identify the recurrent alleles, i.e., mutations demonstrating higher frequency in Slavs than in non-Slavic ethnic communities, and/or identified as of Slavic origin by haplotyping. We were particularly interested in (a) the relative share of a given pathogenic allele among other genetic defects identified in patients with a certain disease, (b) its absolute prevalence in relevant patients from Slavic countries, and (c) its population frequency in Slavic communities. Apart from pathogenic (P) or likely pathogenic variants (LP), we considered 11 variants with conflicting interpretations and 17 variants of uncertain significance, which can be reclassified to P/LP categories according to the ACMG criteria [39]. The catalog of identified variants, their frequency in Slavic and other populations, and data on haplotype evaluation (if any) are presented in Appendix A.

## 4. Distribution of Medically Relevant Alleles in the Slavs

### 4.1. Inborn Errors of Metabolism

The molecular epidemiology of phenylketonuria has been studied with a significant level of comprehension across almost all Slavic populations, thanks to the implementation of neonatal screening for this disorder. The most common cause of phenylketonuria in Eastern Europe is the well-known pan-European variant of Balto-Slavic origin, c.1222C>T (p.Arg408Trp) in the *PAH* gene. The proportion of patients carrying at least one p.Arg408Trp allele is approximately 98% in Estonia, 89% in Poland, 79% in Russia, but only 4% in Spain [40]. It has been suggested that this mutation initially emerged in Lithuania. Interestingly, the same substitution, but in the context of another haplotype, is recurrent in Eastern Ireland (Connacht), indicating that the codon 408 in the *PAH* gene is a mutation hotspot [41]. In Serbia, one-third of *PAH* pathogenic alleles are attributed to the founder c.143T>C (p.Leu48Ser) variant; however, this allele is rare in other populations [40].

The most known pan-European mutation c.3207C>A (p.His1069Gln) in the *ATP7B* gene, which is associated with Wilson’s disease, is also of Balto-Slavic origin and demonstrates a noticeable gradient decreasing from the North-East (72% of pathogenic alleles in Polish patients) to the South-West of Europe (6% in Spain) [42]. Slavic populations are also characterized by a high occurrence of pan-European *FAH* c.554-1G>T (e.g., 42% of pathogenic alleles in Russian patients) and c.1062 + 5G>A (17% of pathogenic alleles in Russian patients) variants, which are associated with the development of hereditary tyrosinemia type I [43], as well as increased frequency of *IDS* (iduronate-2 sulfatase) alleles c.253G>A, c.257C>T, c.263G>A, c.998C>T, c.1327C>T, and c.1403G>A, which are linked to Hunter’s syndrome (mucopolysaccharidosis type 2) [44,45]. 

Pan-European variant *HGD* c.481G>A (p.Gly161Arg) was found in 45 out of 49 Russian patients with alkaptonuria [46]. This allele, together with the c.1278insC (Pro370fs) variant, dominates in Slovakia, the country with the highest prevalence of alkaptonuria (1:19000). Most cases of alkaptonuria occur in one small part of this country (Kosice); as many as five founder variants including a region-specific c.975G>A (Gly270Arg) substitution have been found in this population [47].

The molecular epidemiology of Smith–Lemli–Opitz syndrome (SLOS) has been studied only in Western Slavic populations (Czech Republic and Poland). The most common variant, *DHCR7* c.452G>A (p.Trp151*), demonstrates high frequency in Slavic patients (33–50% in patients from Poland and Czech Republic), while being less frequent in Germany (17%) and the United States (10%) [48]. 

Mucopolysaccharidosis type VI (Maroteaux-Lamy) is associated with several recurrent variants in Slavic populations. Besides the Balto-Slavic pathogenic variant c.454C>T (p.Arg152Trp) in the arylsulfatase gene (*ARSB*), there are also Belarusian c.797A>C (p.Tyr266Ser) and Russian c.1562G>A (p.Cys521Tyr) “regional” *ARSB* mutations contributing to the occurrence of this disease in Slavic populations [49,50]. 

Slavic countries are characterized by a number of recurrent genetic alterations, which affect genes involved in the assembly of the mitochondrial respiratory chain components. The *SURF1* c.845_846delCT mutation, which is rare in other ethnic groups, is a founder allele in Polish (78% of pathogenic alleles in *SURF1* gene) and Russian (65% of pathogenic alleles) patients with Leigh syndrome [51,52]. The c.418G>A (p.Glu140Lys) mutation in the *SCO2* gene, being the most common in the Czech Republic (83%) and Poland (86%), is associated with the development of mitochondrial encephalocardiomyopathy in childhood [53]. This allele has also been repeatedly found in the exomes of Russian individuals [54]; however, the incidence of *SCO2*-related diseases in Russia is still unknown. The c.3G>A (p.Met1?) and c.494A>T (p.Glu165Val) variants in the *DGUOK* gene, which are associated with mitochondrial DNA depletion syndrome, are overrepresented in Polish (43–70% and 29% of pathogenic alleles, respectively) [55,56] and Russian patients (c.3G>A: 91% of pathogenic variants) [57]. A recurrent variant in the *TWNK* gene, c.1199G>T (p.Arg400Leu), is probably specific to the Russian population [57].

*MMAA* c.593_596del (p.Thr198fs) accounts for three-quarters of pathogenic alleles identified in Polish patients with a vitamin B12-dependent methylmalonic aciduria [58]. The high frequency of the *ADSL* variant c.1277G>A (p.Arg426His) has been demonstrated in Polish (50% of pathogenic alleles) and Czech (30%) patients with adenylosuccinate lyase deficiency; this allele is also often found among German patients but is not characteristic for other Slavic populations [59,60].

*LIPA* c.894G>A substitution is the most common allele in Russian [61] and Polish [62] patients with lysosomal acid lipase deficiency, although it also occurs in other populations. In addition, Russian patients with this disease often carry a unique allele c.420G>A (p.Trp140*) [61].

Pan-European mutation *GCDH* c.1204C>T (p.Arg402Trp) has a particularly elevated occurrence in Slavic patients with glutaric aciduria type 1 [63,64]. The second most common variant detected in Russian patients is c.1262C>T (p.Ala421Val); this allele was initially described in Amish communes located in Pennsylvania [65].

Two unrelated Russian patients, who suffered from hyperammonemia due to carbonic anhydrase VA deficiency, have been reported recently; both these subjects were homozygous for *CA5A* c.555G>A p.(Lys185=) allele [66]. This splicing-affecting substitution leads to exon 4 skipping and results in p.Leu154_Lys185del. This variant has minor allele frequency (MAF) = 0.0025 in Russians, which is the highest frequency worldwide.

Alpha-mannosidosis is frequently associated with *MAN2B1* c.2248C>T (p.Arg750Trp, also known as p.Arg749Trp) recurrent variant in European patients (roughly 25% of pathogenic alleles). The impact of this substitution is significantly more pronounced in Polish patients (60% of pathogenic alleles) [67]. Moreover, it demonstrates the highest MAF in populations of Balto-Slavic ancestry (0.003 in Estonians, 0.002 in Russians, and 0.00075 in Bulgarians) (Appendix A).

Kashubians are characterized by an exceptionally high frequency of pathogenic *HADHA* c.1528G>C (p.Glu510Gln) allele (carrier frequency up to 1:57), which is associated with severe recessive encephalocardiomyopathy and HELLP syndrome in heterozygous carriers pregnant with a fetus with biallelic *HADHA* inactivation [68].

### 4.2. Hereditary Cancer Syndromes

Molecular epidemiology of *BRCA1/2*-associated hereditary breast and ovarian cancer (HBOC) has been comprehensively characterized across almost all Slavic populations. *BRCA1* pathogenic variants are generally more prevalent than *BRCA2* defects in Eastern and Western Slavic HBOC patients. Slavic *BRCA1* c.5266dupC (p.Gln1756ProfsX74, also known as 5382insC) allele is particularly common in populations of Eastern European descent [69]. The share of this allele among all pathogenic *BRCA1* mutations approaches 50–70% in Russian, Polish, and Czech patients. Southern Slavs demonstrate a steady decrease in c.5266dupC allele frequency from North-East to South-West; this variant has a null frequency in Croatia and Slovenia (Appendix A and [69]).

*BRCA1* c.181T>G (p.Cys61Gly) is characteristic for Central Europe and is particularly common in Slovenia (25%), North Macedonia (23%), Poland (23%), Belarus, and Ukraine (Appendix A). Poland (4%) and Russia (7.5%) demonstrate a noticeable frequency of the “Lithuanian” founder allele, *BRCA1* c.4035delA (p.Glu1346fs). Poland and the Czech Republic also show a high frequency of *BRCA2* c.658_659del (p.Val220Ilefs*4) variant (6% and 3%, respectively; Appendix A; [70]). There are several “minor” genetic alterations, which are either typical for almost all Slavic countries (*BRCA1* c.3695_3699GTAAA (p.Val1234fs)), or certain areas populated by Slavic people (Western Slavic *BRCA2* c.9403del (p.Leu3135Phefs*28) variant), or individual Slavic countries (e.g., *BRCA2* c.8537_8538del (p.Glu2846Glyfs*22) in Czech Republic, see also Appendix A).

*BRCA2* c.5286T>G (p.Tyr1762*) is a Northern Russian founder variant accounting for half of *BRCA2* pathogenic alleles in hereditary breast-ovarian cancer patients from the Arkhangelsk region [71]. Regional clustering of founder alleles is very uncommon for Slavic inhabitants of Russia, despite the large territory of this country. Interestingly, population genetic studies have suggested that autochthonous inhabitants of several sparsely populated Northern Russian territories (including the Arkhangelsk region), being Russians by self-determination, are genetically closer to Finno-Ugric ethnic groups than to Slavs [12]. 

Southern Slavs have a few *BRCA1/2* founder mutations usually characteristic for some but not all countries in this region. In addition, there is an unexpected regional enrichment for several minor European alleles (Appendix A).

*PALB2* truncating mutations are associated with a high risk of breast and possibly some other cancers. There are two Slavic founder alleles: c.168_171delTTGT (p.Gln60fs, also known as c.172-175delTTGT) and c.509_510delGA (p.Arg170Ilefs); the latter variant is also characteristic for German patients [72,73,74]. Russia, Poland, and Czech Republic have a high prevalence of recurrent *CHEK2* cancer-predisposing mutations represented by a “Scandinavian” allele c.1100del (p.Thr367fs) and two Slavic variants (c.444 + 1G>A and c.(908 + 1_909-1)_(1095 + 1_1096-1)del) [74,75,76]. The combined carrier frequency of recurrent *CHEK2* variants exceeds 1% in Russia and Poland. *CHEK2* mutations are associated with a moderate risk of breast, kidney, thyroid, and testicular cancer, and, possibly, some other neoplasms [77,78]. Two truncating alleles, c.1667_1667 + 3delAGTA in *RECQL* [74,79,80] and c.1152_1155del (p.Thr384_Gly385insTer) in *ATRIP* [81], being recurrent in Poland, Belarus, and some other Central/Eastern European countries, are likely to be associated with a low-penetrance predisposition to breast cancer. *RECQL* c.1667_1667 + 3delAGTA variant has the highest MAF (0.001076) in the Swedish subset of the gnomAD NFE cohort, while *ATRIP* c.1152_1155del allele reaches its highest frequency in Slavic populations (Appendix A). 

The second most common genetic tumor syndrome, hereditary non-polyposis colon cancer, does not show a strong founder effect in Slavic populations. Nevertheless, a minor mutation c.677G>T (p.Arg226Leu) in the *MLH1* gene has been repeatedly found in Poland, Russia, and Slovakia; several Lynch syndrome-associated variants are very common in North Macedonia (up to 40–60% of all pathogenic alleles) [82,83].

Polish patients with paragangliomas often carry *SDHD* c.33C>A (p.Cys11*) variant; all subjects with this allele have identical haplotypes indicating a founder effect [84]. *SDHD* c.305A>G (p.His102Arg) substitution is the most common allele in Russian paraganglioma patients [85].

A small Russian study recently showed the recurrent status of *CDKN2A* c.307_308del (p.Arg103fs) allele: it was identified in three out of six (50%) patients with hereditary cancer syndrome manifesting by a predisposition to melanoma and pancreatic malignancies [86]. This mutation is exceptionally rare outside Russia.

### 4.3. Neurological and Neuromuscular Diseases

Molecular epidemiology of spinocerebellar ataxia (SCA) has been systematically studied in Russia and Poland. Surprisingly, up to 75% of instances of this disease in Slavs are attributed to alterations of the *SCA1* gene, which are relatively uncommon in other parts of the world. It is of notice that mutations in the *SCA3* gene, being the most frequent cause of SCA worldwide, show limited contribution to this disease in Slavic countries [87,88,89]. The unusually high prevalence of CAG repeat expansion in *SCA1* seems to be significantly influenced by a founder effect [90].

Alterations of the *CAPN3* gene are associated with limb-girdle muscular dystrophy type 2A. The *CAPN3* allele c.550delA (p.Thr184ArgfsX36), which is common in Central Europe, reaches the highest frequency in Slavic populations (more than 70% of all pathogenic alleles in Serbian and Croatian patients) [91,92,93].

A few Slavic alleles are associated with various rare neurological diseases. For example, the *GDAP1* c.715C>T (p.Leu239Phe) variant has been found in Polish and Russian patients with Charcot–Marie–Tooth disease, type 4A [94,95]. *HINT1* c.110G>C (p.Arg37Pro) causes neuromyotonia with axonal neuropathy; this allele is particularly common in Czech (95%) and Russian (97%) patients [96,97]. A recent study showed a slightly lower prevalence of this variant in Lithuanian patients (75%) [98]. All Polish subjects with hereditary spastic paraplegia, type 47 carry the same *AP4B1* c.1160_1161del (p.Thr387fs) mutation [99].

Defects in the *C19orf12* gene are associated with neurodegeneration with brain iron accumulation (NBIA4). *C19orf12* c.204_214del (p.Gly69ArgfsX10) allele is repeatedly found in patients of Polish and other Eastern European ancestry [100,101,102].

Recurrent mutations *LAMA2* c.799G>A (p.Asp267Asn) and c.9095dupA (p.Ile3033Aspfs*6) have been identified in Czech patients with muscular dystrophy [103]. Russian patients do not carry these alleles, but demonstrate elevated occurrence of another pathogenic variant, *LAMA2* c.7536delC (p.Asp2513fs) (21% of all pathogenic alleles) [104].

A recent study revealed a number of recurrent alleles in Russian patients with *GNE*-associated myopathy. One of these variants, *GNE* c.1760T>C (p.Leu587Ser), has only once been reported outside Russia [105].

### 4.4. Hereditary Endocrinopathies

Germ-line defects in the *PROP1* gene are associated with a combined pituitary deficiency. The two most common European pathogenic variants, c.150del (p.Arg53Aspfs) and c.301_302delGA (p.Leu102Cysfs), have Balto-Slavic origin [106]. The distribution of these alleles in European countries demonstrates a gradient from the North-East to the South-West; the c.301_302delGA variant is also characterized by elevated occurrence in Spain and Portugal. Haplotyping of carriers of this allele showed that this mutation has independently emerged at least twice in a resident of the Iberian Peninsula and in a subject of Balto-Slavic origin. There are carriers of both the “Iberian” and “Balto-Slavic” p.Leu102Cysfs allele in the countries of the New World [106]. A regional founder mutation c.150_151del (p.Gly52fs) has also been described in Poland and Russia [107].

The *GNRHR* gene is associated with non-syndromic hypogonadotropic hypogonadism. Slavic populations demonstrate a high frequency of *GNRHR* c.416G>A (p.Arg139His) allele (up to 0.0019 MAF in Bulgarians). This allele is also often found in Brazil, but the Brazilian and Polish haplotypes turned out to be different; most likely, they appeared independently [108]. 

*TPO* c.1430_1450del (p.Ala477_Asn483del) allele is associated with familial thyroid dyshormonogenesis 2A. It is common in Slovenia, Bosnia, and Slovakia (16% of *TPO* pathogenic alleles), probably representing a regional founder effect [109].

Slavic mutation c.787T>C (p.Ser263Pro) in the *AAAS* gene, which is linked to Allgrove syndrome (AAA syndrome: achalasia, Addison disease (adrenal insufficiency), and alacrimia), was detected mainly in subjects from Croatia, Poland, Czech Republic, and Slovenia [110,111]. c.1159C>T (p.Gln387*) is another recurrent *AAAS* allele, which has been described in Croatian patients [111].

### 4.5. Inborn Errors of Immunity

Nijmegen syndrome is largely a Slavic disease, being exceptionally rare in non-Slavs. Almost all cases of this disease are associated with the *NBN* c.657_661del (p.Lys219fs) mutation demonstrating high frequency in most Slavic populations (carrier frequency 0.5–1.0%) [112,113] (Figure 2).

*C2* c.841_849 + 19del (p.Val281fs) allele is common in individuals of European descent. Homozygosity or heterozygosity for inactivating mutations in this gene is associated with low CH50 complement activity and entails a predisposition to recurrent infections and autoimmune diseases. This allele was repeatedly detected among Slovenian patients with complement deficiency [114]; according to the data obtained from exome studies, it is common in Slavic and Baltic countries, with MAF value approaching 0.02 [54].

A defect of another component of the complement system, *C8B*, is associated with a highly elevated risk of generalized meningococcal infection. *C8B* mutation c.1282C>T (p.Arg428*) is often found in patients from Russia (90% of pathogenic alleles) and Slovenia (100% of pathogenic alleles) [114,115].

The c.256_257delAA (p.Lys86Valfs*33) variant in the *RAG1* gene can also be considered a classical Slavic mutation; homozygosity for this position is clinically associated with severe combined immunodeficiency (SCID) phenotype [116,117]. A rarer form of SCID is caused by *RAG2* deficiency. There are recurrent *RAG2* alleles in Russia (c.1300T>C (p.Tyr434His)) and Poland (c.1357T>C (p.Trp453Arg)) [116].

Russian patients with hemophagocytic lymphohistiocytosis often carry the *UNC13D* c.3037insG (p.Asp1013Glyfs*11) pathogenic variant [118].

The well-known Finnish mutation *AIRE* c.769C>T (p.Arg257*), which is associated with Autoimmune Polyendocrinopathy–Candidiasis–Ectodermal Dystrophy syndrome (APECED), has been reported as the most common cause of this disease in Poles (71% of pathogenic alleles), Russians (70%), Slovenes (70%), and Serbs (92%) [119,120].

### 4.6. Hereditary Kidney Diseases

Hypomorphic mutation c.1871G>A (p. Gly624Asp) in the *COL4A5* gene is detected in a high proportion of patients with X-linked dominant Alport syndrome (XLD-AS) residing in Southern and Central Europe. Most often this mutation occurs among XLD-AS patients from Slovenia (35%), Cyprus (33%), and Hungary (30%); it is also found in patients from Russia (16%) and Australia (11%) [121,122,123,124,125,126]. This is a frequent cause of adult-onset chronic renal failure in these populations. A recent study from Poland provides evidence that all c.1871G>A alleles share the same rare haplotype, dating the origin of this mutation back to the 12–13th centuries [127]. The phenotype of this disease is intermediate between the so-called basement membrane disease (BMTD), which is a benign condition, and severe Alport syndrome (early-onset and rapidly progressing glomerular disease, with hearing loss and ocular abnormalities). While 90% of subjects with conventional Alport syndrome experience end-stage renal disease (ESRD) by the age of 30 years, approximately half of patients with the *COL4A5* c.1871G>A variant still do not require dialysis in their fifties, and only a quarter of male carriers of this allele suffer from hearing loss [127]. 

The *NPHS2* c.868G>A (p.Val290Met) variant is recurrent in Czech patients with steroid-resistant nephrotic syndrome (SRNS); it accounts for approximately 75% of all *NPHS2* pathogenic alleles [128]. It is also common in Polish patients (20% of *NPHS2* disease-causing variants) [129]. This substitution has previously been considered to be a minor Central European allele [130]. Carriers of the *NPHS2* c.868G>A variant share the same haplotype [128]. *NPHS2* c.868G>A allele demonstrates relatively high MAF in Russians (0.0025), Estonians (0.001), and Bulgarians (0.00075), compared to the rest of the European populations. The fact that it has not been previously recognized as a frequent cause of SRNS in the Czech Republic and other Slavic countries is probably linked to the hypomorphic nature of the *NPHS2* c.868G>A p.Val290Met mutation. While typical *NPHS2*-associated SRNS, which is caused by fully inactivating *NPHS2* pathogenic variants, generally manifests in childhood, *NPHS2* c.868G>A (p.Val290Met) allele carriers usually develop the disease in adult age, and, therefore, are rarely subjected to genetic testing [128,130]. 

A rare form of inherited steroid-resistant nephrotic syndrome, which is associated with the founder c.1772G>T (p.Gly591Val) mutation in the *NUP93* gene, was identified in 21 patients (including 10 homozygotes) from Central and Eastern Europe (Poland, Czech, Germany, Hungary, Russia, Serbia) and Turkey. All carriers of this substitution share the same haplotype [131]. 

The *SLC7A9* c.313G>A (p.Gly105Arg) mutation is a common hereditary cause of kidney stones in the Southern Slavic countries (11–50%) and occurs at some frequency in other regions of Southern Europe and Balkans, as well as in Turkey [132].

Some tubulopathies, accompanied by nephrocalcinosis, are characteristic of Slavic countries and associated with founder alleles. For example, *CYP24A1* c.1186C>T (p.Arg396Trp) substitution represents at least half of *CYP24A1* pathogenic alleles in Russia and Poland [133,134].

Germ-line alterations in the *CLDN16* gene are associated with renal hypomagnesemia type 3. Recurrent *CLDN16* c.453G>T (p.Leu151Phe) allele is often found in Poland (72% of pathogenic alleles), Serbia (100%), and other Slavic countries, as well as in Germany (48%) [135,136,137]. 

Founder variants *FRAS1* c.6963_6964dup (p.Val2322fs), associated with Fraser’s syndrome, and *ALMS1* c.11880_11881delTT (p.Ser3961Leufs*11), linked with Alstrom’s syndrome, have been identified in Poland [18,138]. Kashubians, a Slavic ethnic isolate living in the North-West of modern Poland, are characterized by the persistence of a “regional” founder variant *NPHS2* c.1032delT (p.Phe344Leufs*5) causing steroid-resistant nephrotic syndrome [139].

### 4.7. Hearing Loss and Eye Diseases

A classic example of a pan-European mutation is the *GJB2* c.35delG variant, which is a major hereditary cause of hearing loss across European populations [140]; among countries with predominantly Slavic populations, the highest frequency of this pathogenic allele was found in Belarus, where its carrier frequency reaches 5.7% [141,142]. 

The second most common genetic cause of recessive deafness in Poland, Belarus, and Russia is the Balto-Slavic *GJB2* c.313_326del (p.Lys105fs) allele, presumably of Lithuanian origin [143,144,145]. Slovak patients commonly have the c.71G>A (p.Trp24*) pathogenic variant, which is frequent in South Asian populations; it was revealed that Slovak carriers of this mutation have Gypsy ethnic roots [146]. Except for the relatively high frequency of two Scandinavian variants in the *TMPRSS3* gene found in Polish patients with hearing impairment [147], recurrent causes of GJB2-negative hearing loss in Slavic populations remain largely unknown.

The *CHST6* c.599T>G (p.Leu200Arg) variant demonstrates an unusually high representation among Polish (44%) and Czech (58%) patients with macular corneal dystrophy [148,149]. A recently discovered autosomal recessive form of Leber’s optic neuropathy is linked to the *DNAJC30* c.152A>G (p.Tyr51Cys) variant in patients from Russia, Poland, and Ukraine [150].

There are several Slavic founder-mutations associated with Usher’s syndrome. *USH2A* c.11864G>A (p.Trp3955*) mutation is often detected in patients from Central and Southern Europe. Its share in the spectrum of pathogenic *USH2A* alleles reaches 83% in Slovenia, also being high in the Czech Republic (53%) and Russia (30%) [151,152,153]. Slovenians also have a minor regional founder *USH2A* variant, c.2610C>A (p.Cys870*), which is extremely rare outside this country [151]. The c.52C>T (p.Gln18*) variant in the *MYO7A* gene, another gene for Usher’s syndrome, was described in Slovenian (6/12, 50%) and Russian patients (3/14, 21%) [151,153].

### 4.8. Skin Diseases

The pan-European *TGM5* variant c.337G>T (p.Gly113Cys), which is associated with acral peeling syndrome, is recurrently detected in Poland, the Czech Republic, and Russia [54,154,155,156,157].

A series of systematic studies of hereditary genodermatoses was carried out in the Czech Republic. In addition to common pan-European alleles, these investigations revealed an unusually high frequency of several recurrent mutations associated with autosomal recessive ichthyosis and epidermolysis bullosa (*ALOXE3* c.1096C>T (p.Arg366*): 50% of the pathogenic alleles; *CYP4F22* c.59dupG (p.Ile21Hisfs*59): 50%; *COL7A1* c.425A>G (p.Lys142Arg): 16–30%; c.6146G>A (p.Gly2049Glu): 8-9%) [155,158,159,160]. The latter *COL7A1* variant is confined to the Czech Republic, while the former is considered to be a Central European founder mutation. Its highest prevalence is observed in Poland (37% of all pathogenic alleles) [161], while in Germany and Hungary this estimate is approximately 10–13% [157,161]. In addition, a minor recurrent variant *COL7A1* c.682 + 1G>A, which is common in Polish (9%), Russian (6%), and Czech (4%) patients, has been described [157,158,161]. 

A very rare palmoplantar keratoderma, called Meleda Island disease or Mal de Meleda, occurs mainly in some small regions in Turkey, Algeria, and the Croatian island of Mljet. Molecular genetic studies revealed two recurrent pathogenic alleles in the *SLURP1* gene [162,163]. One of the alleles, c.82delT (p.C28fs32*), is common both in Croatian (57%) and Algerian (75%) patients [162], while the other, c. 286C>T (p.Arg96*), is found exclusively in Croatian families (43%) [163].

### 4.9. Heart, Blood, and Lung Diseases

Autosomal dominant familial hypercholesterolemia, being the most common hereditary heart disorder, generally does not demonstrate a pronounced founder effect. However, the *LDLR* c.1775G>A (p.Gly592Glu) variant accounts for up to 9–22% pathogenic alleles in patients from Russia, Poland, Slovakia, and the Czech Republic [164,165,166,167]. The duplication of exons 4–8 in the *LDLR* gene is recurrent in Polish patients [164]. *LDLR* c.662A>G (p.Asp221Gly) allele is particularly frequent in Kashubia [168].

The recurrent *MYBPC3* c.2541C>G (p.Tyr847*) variant is associated with autosomal-dominant hypertrophic cardiomyopathy (AD HCM) in Poland [169]. This alteration was also described in a patient from Russia [170]. Still, a recent study of a large cohort of Russian HCM patients demonstrated the high prevalence of another *MYBPC3* variant, c.3697C>T (p.Gln1233*) (27% of all pathogenic alleles) [171]. The *MYBPC3* c.3697C>T (p.Gln1233*) mutation was also frequently detected in Czech (23%) and Hungarian (22%) patients, but was not observed in Polish or Slovak HCM studies [172,173]. It is not known, whether Russian, Czech, and Hungarian *MYBPC3* c.3697C>T carriers share the same haplotype. There is also a recurrent *TPM* c.629A>G (p.Gln210Arg) variant, which accounts for 50% of disease-associated alleles in Russian HCM patients [171]. The *FHOD3* c.1646 + 2T>C mutation is the second most common genetic cause of AD HCM in Slovenian patients reporting Serbian, Bosniak, or Montenegrin roots. It accounts for 100% of all pathogenic *FHOD3* alleles and 16% of all genetically verified HCM cases in this category of subjects, while being exceptionally rare in other populations [174].

There are several regional Slavic founder mutations associated with coagulopathies. The *SERPINC1* c.1157T>C (p.Ile386Thr) variant is detected in 16% of antithrombin deficiency type III patients in Poland [175]. It has previously been identified in several patients of Central European descent. *F8* c.1901A>G (p.His634Arg) allele is endemic for the population of Sverdlovsk Region (Russia); it accounts for approximately 25% of hemophilia A cases observed in residents of this area [176].

### 4.10. Other Hereditary Diseases

A study of Polish patients with primary ciliary dyskinesia (PCD) has resulted in the identification of two recurrent alleles, *DNAI1* c.1612G>A (p.Ala538Thr) and *ZMYND10* c.367delC (p.His123Thrfs*16) [177,178].

The transthyretin *TTR* c.325G>C (p.Glu109Gln, also known as p.Glu89Gln) mutation, which causes a hereditary type of amyloidosis, is common in many Mediterranean countries. The highest percentage of this allele (75%) is detected among Bulgarian patients, all of whom have the same haplotype [179].

Pan-European pathogenic alleles are almost always present in Slavic countries of Eastern Europe. However, the recurrent European alteration, *CFTR* p.1521_1523delCTT (p.Phe508del), accounting for the majority of pathogenic alleles in Western European patients with cystic fibrosis, is substantially less common in Slavic populations: its occurrence approaches about 50–60% in patients from Russia, Poland, and Slovakia [180]. Among minor *CFTR* alleles, there is a large intragenic deletion c.54-5940_273 + 10250del21kb (CFTRdele2,3), which accounts for 1–6% of pathogenic variants in Poland and Russia [181,182,183]. Some of the minor *CFTR* alleles reflect the presence of regional founder effect, e.g., c.2052dupA (p.Gln685Thrfs) characteristic for Galicia (Western Ukraine) [184].

In Slovakia and, to a lesser extent, in Poland, there is substantial occurrence of *PRNP*-associated prion encephalopathy, a condition that is extremely rare in other countries. This is attributed to the persistence of a recurrent pathogenic substitution *PRNP* c.598G>A (p.Glu200Lys) [185,186]. This variant also occurs in non-Slavic populations (Chile, Libyan, and Tunisian Jews); however, haplotyping studies suggest the existence of several ancestors. 

Congenital chloride diarrhea occurs in Finland, some Arab countries, and Poland, but is extremely rare in other regions. In Poland, this condition is associated with the c.2024_2026dup (p.Ile675dup) allele of the *SLC26A3* gene [187].

The examples of the most frequent recurrent pathogenic alleles described in Slavic communities are listed in Table 2. It is necessary to acknowledge that many of these variants demonstrate huge interstudy variations with regard to their frequency. Differences in ethnic or geographical origin of the patients, small study size, selection bias, or technical limitations may substantially contribute to these inconsistencies; however, the analysis of the involved confounding factors is beyond the scope of this review.

**Table 2 ijms-25-09335-t002:** The most frequent recurrent pathogenic variants described in Slavic populations.

Variant	Associated Disease	Comments	References
A. Pan-European alleles demonstrating increased frequency in Slavic populations ^1^			
*AIRE* c.769C>T (p.Arg257*)	Autoimmune polyendocrinopathy–candidiasis–ectodermal dystrophy (APECED)	Major founder mutation, frequent in Finnish and Slavic populations MAF: 0.0030 (Russians), 0.00116 (Bulgarians) High share of pathogenic alleles in Serbian (92%), Polish (71%), Slovenian (70%), and Russian (70%) patients	[119,120,188,189]
*ATP7B* c.3207C>A (p.His1069Gln)	Wilson’s disease	Major mutation in Central and Eastern Europe, most probably of Balto-Slavic origin. MAF: 0.0061 (Russians), 0.00263 (Bulgarians) High share of pathogenic alleles in Czech (72%), Polish (66%), Bulgarian (62%), Slovakian (56%), Croatian (54%), and Russian (51%) patients	[42,190,191]
*BRCA1* c.5266dupC (p.Gln1756ProfsX74), also known as 5382insC	Hereditary breast and ovarian cancer (autosomal dominant inheritance)	Pan-Slavic mutation of Russian or, possibly, Danish origin, predominant in Eastern and Western Slavic populations and less frequent in Southern Slavs MAF: 0.0016 (Russians), 0.00075 (Bulgarians) High share of pathogenic alleles in Russian (68%), Polish (55%), Belarusian (52%), Bulgarian (56%), Serbian (37%), Czech (32%), and Slovakian (25%) patients	[69,74,192,193,194]
*BRCA1* c.181T>G (Cys61Gly), also known as 300T>G		Major founder mutation in Central and Eastern Europe MAF: 0.0004 (Poles), 0.00037 (Bulgarians), 0.0001 (Russians) High share of pathogenic alleles in Polish (23%), Macedonian (23%), and Slovenian (25%) patients	
*BRCA1* c.4035delA (p.Glu1346fs), also known as 4153delA		Major founder mutation in Baltic States, Belarus, Poland, Russia (most probably of Lithuanian origin) MAF: 0.00013 (Poles), 0.0001 (Russians) Noticeable share of pathogenic alleles in Polish (4%), Russian (8%), and Belarusian (16–33%) patients	
*C2* c.841_849 + 19del (p.Val281Profs)	C2 complement deficiency	MAF: 0.0189 (Russians), 0.025 (Poles) High share of pathogenic alleles in Slovenian patients (85%)	[54,114,195]
*C8B* c.1282C>T (p.Arg428*)	Complement component 8B deficiency	MAF: 0.0071 (Russians), 0.0075 (Bulgarians) High share of pathogenic alleles in Russian (90%) and Slovenian (100%) patients	[114,115]
*CFTR* c.1521_1523delCTT (p.Phe508del)	Cystic fibrosis	Major founder mutation of Western European origin, highly prevalent in all Slavic populations examined (Poles, Bulgarians, Russians, Czechs, Slovaks)	[180,183,196]
*CYP24A1* c.1186C>T (p.Arg396Trp)	Idiopathic infantile hypercalcemia	MAF: 0.0058 (Russians), 0.00449 (Bulgarians) High share of pathogenic alleles in Polish (61%) and Russian (50%) patients	[133,134]
*PAH* c.1222C>T (p.Arg408Trp)	Phenylketonuria	Recurrent mutation independently originated in Balto-Slavic and Irish populations, common in Eastern and Western Slavs (Poles, Russians, Czechs, Slovaks) MAF: 0.0091 (Russians), 0.00187 (Bulgarians) High share of pathogenic alleles in Polish (62%), Ukrainian (52%), Russian (51%), Slovakian (49%), Czech (42%), Slovenian (28%), and Serbian (16%) patients	[40,197,198]
*PROP1* c.301_302delGA (p.Leu102Cysfs)	Combined pituitary hormone deficiency-2	Recurrent mutation independently originated in Balto-Slavic and Spanish/Portugal populations MAF: 0.00502 (Russians), 0.00113 (Bulgarians) High share of pathogenic alleles in Polish (88%), Russian (79%), and Czech (72%) patients	[106,107]
*IDUA* c.208C>T (p.Gln70*)	Mucopolysaccharidosis type I	Major founder mutation of Northern European origin, highly prevalent in Eastern and Western Slavs (Russians, Poles, Czechs, Slovaks)	[199,200,201,202]
*TGM5* c.337G>T (p.Gly113Cys)	Acral peeling skin syndrome	Major founder pan-European mutation, most frequent in Slavic populations MAF: 0.0062 (Russians), 0.00562 (Bulgarians) High share of pathogenic alleles in Czech (93%) and Polish (72%) patients	[154,155]
B. “Pan-Slavic” alleles shared by at least two Slavic communities, but infrequently or never reported in most other populations ^1^			
*ARSB* c.454C>T (p.Arg152Trp)	Mucopolysaccharidosis type VI	Balto-Slavic mutation, predominant in Eastern and Western Slavs (Russians, Poles, Belarusians)	[49,50,202]
*ATRIP* c.1152_1155del p.Thr384_Gly385insTer	Low-penetrance breast cancer predisposition (autosomal dominant inheritance)	This mutation is much more frequent in Balts and Slavs than in other European populations MAF: 0.0007 (Russians), 0.00059 (Poles), 0.00037 (Bulgarians)	[81]
* ATM * c.5932G>T (p.Glu1978*)	Ataxia-telangiectasia	Slavic mutation, predominant in Russia and common in Poland and Belarus MAF: 0.0007 (Russians), 0.0005 (Belarusians), 0.00037 (Bulgarians), 0.00025 (Poles) Noticeable share of pathogenic alleles in Polish (11–16%) and Russian (43%) patients	[203,204,205,206]
*BLM* c.1642C>T (p.Gln548*)	Bloom syndrome Low-penetrance breast cancer predisposition in heterozygotes?	Slavic mutation, prevalent in Russians, Belarusians, Slovaks, Czechs MAF: 0.0023 (Russians), 0.00189 (Bulgarians), 0.00275 (Poles), 0.001 (Belarusians) High share of pathogenic alleles in Polish, Russian, Slovak, and Czech patients	[74,207,208,209]
*CHEK2* c.(908 + 1_909-1)_(1095 + 1_1096-1)del	Low-penetrance breast, kidney, thyroid, etc., cancer predisposition (autosomal dominant inheritance)	Slavic founder mutation MAF: 0.002 (Poles) Noticeable share of pathogenic alleles in Polish (38%), Czech (30%), and Slovenian (12%) patients	[76,210,211]
*CHEK2* c.444 + 1G>A		Slavic founder mutation MAF: 0.001 (Russians), 0.00037 (Bulgarians) Noticeable share of pathogenic alleles in Polish (35%), Slovenian (31%), and Czech (8%) patients	[74,76,211]
*CAPN3* c.550delA (p.Thr184Argfs)	Limb-girdle muscular dystrophy type 2A	Slavic mutation. Also frequent in North-Eastern Italy, Turkey, Germany MAF: 0.0027 (Russians), 0.004 (Poles) High share of pathogenic alleles in Croatian (74%), Serbian (71%), Bulgarian (59%), Russian (53%), Polish (52%), Slovakian (49%), and Czech (38%) patients	[92,93,212,213,214]
* CFTR * c.54-5940_273 + 10250del21kb (CFTRdele2,3)	Cystic fibrosis	Slavic mutation, highly prevalent in Czechs, Russians, Ukrainians, and Poles	[180,181,182]
* COL4A5 * c.1871G>A (p.Gly624Asp)	X-linked Alport syndrome X-linked dominant (hypomorphic allele)	Slavic and Balkan mutation (also frequent in Hungary and Cyprus). MAF: 0.0012 (Russians), 0.0005 (Bulgarians) High share of pathogenic alleles in Polish (39%), Slovenian (35%), and Russian (16%) patients	[122,124,125,127]
*DHCR7* c.452G>A (p.Trp151*)	Smith–Lemli–Opitz syndrome	Slavic mutation, also relatively frequent in Germany (17%) and the USA (10%) MAF: 0.0057 (Russians), 0.00451 (Bulgarians) High share of pathogenic alleles in Czech (47%) and Polish (33%) patients	[48,215]
* DNAJC30 * c.152A>G (p.Tyr51Cys)	Leber hereditary optic neuropathy, autosomal recessive	Slavic mutation, highly prevalent in Russians, Poles, Ukrainians MAF 0.0084 (Russians), 0.004 (Bulgarians) All pathogenic alleles in Russian (100%), Czech (100%), Polish (100%), and Ukrainian (100%) patients	[150]
* GDAP1 * c.715C>T (p.Leu239Phe)	Charcot–Mari–Tooth type 4 disease	Slavic mutation, highly prevalent in Russians, Poles, Czechs	[94,95,156,216]
* HINT1 * c.110G>C (p.Arg37Pro)	Neuromyotonia and axonal neuropathy, autosomal recessive	Balto-Slavic mutation, highly prevalent in Russians and Czechs (no data on other Slavic populations) MAF: 0.0031 (Russians), 0.00375 (Bulgarians), 0.00312 (Estonians) Almost all pathogenic alleles in Russian (97%) and Czech (95%) patients	[96,97]
*LDLR* c.1775G>A p.(Gly592Glu)	Familial hypercholesterinemia (autosomal dominant inheritance)	Slavic or Southern European mutation (frequently occurs in Northern Greece, Andalusia, Italy, and Portugal) MAF: 0.0003 (Russians), 0.00037 (Bulgarians) Noticeable share of pathogenic alleles in Polish (22%), Czech (19%), Russian (17%), and Slovakian (13%) patients	[164,165,166,167,217,218,219]
*MYBPC3* c.3697C>T (p.Gln1233*)	Hypertrophic cardiomyopathy (autosomal dominant inheritance)	Slavic or Central European mutation (frequently occurs in Hungary; occasionally found in some non-Slavic countries) MAF: 0.0003 (Russians), 0.00037 (Bulgarians) Noticeable share of pathogenic alleles in Polish (27%) and Czech (23%) patients	[171,172,173]
*NBN* c.657_661del (p.Lys219fs)	Nijmegen breakage syndrome	Slavic mutation, highly prevalent in Western Slavs (Czechs, Poles, Sorbs), Eastern Slavs (Belarusians, Ukrainians, Russians), and at least some Southern Slavs (Bulgarians) MAF: 0.0033 (Russians), 0.00188 (Bulgarians)	[112,113]
*PALB2* c.168_171delTTGT (p.Gln60fs)	Hereditary breast cancer (autosomal dominant inheritance)	Slavic allele MAF: 0.0012 (Poles) Noticeable share of pathogenic alleles in Polish (31%), Czech (30%), and Russian (8%) patients	[72,74,220]
*PALB2* c.509_510delGA (p.Arg170Ilefs)		Slavic allele, but is also common in Germany MAF: 0.00038 (Poles), 0.0001 (Russians) High share of pathogenic alleles in Polish (52%), Russian (42%), and Czech (8%) patients	[72,74,220]
* RAG1 * c.256_257delAA (p.Lys86fs)	Omenn syndrome/Severe combined Immunodeficiency	Slavic or Balto-Slavic allele, most prevalent in Poles and other Western Slavs, less frequent in Southern Slavs; less common, but still a major mutation in Eastern Slavs	[116]
C. “Regional” Slavic pathogenic alleles, infrequently or never reported in other populations (selected examples)			
*ARSB* c.1562G>A (p.Cys521Tyr)	Mucopolysaccharidosis type VI	Russian mutation	[49,50]
*MLH1* c.392C>G (p.Ser131*)	Lynch syndrome (Hereditary non-polyposis colorectal cancer)	Macedonian mutation	[82]
*NPHS2* c.1032delT (p.Phe344Leufs)	Nephrotic syndrome type 2	Polish (Kashubian) mutation	[139]
*PAH* c.143T>C (p.Leu48Ser)	Phenylketonuria	Serbian mutation	[221]
*SLURP1* c. 286C>T (p.Arg96*)	Mal de Meleda (Mljet disease)	Croatian (Mljet island) mutation	[162]
*SDHD* c.33C>A (p.Cys11*)	Hereditary paraganglioma-pheochromocytoma syndrome	Polish mutation	[84]
*TWNK* c.1199G>T (p.Arg400Leu)	Mitochondrial DNA depletion syndrome	Russian mutation	[57]
*GCDH* c.1262C>T (p.Ala421Val)	Glutaric aciduria, type I	Russian mutation	[63]
*BRCA2* c.5286T>G (p.Tyr1762*)	Hereditary breast and ovarian cancer (autosomal dominant inheritance)	Northern Russian mutation	[71]

^1^ Recessive variants with MAF > 0.003 in any Slavic population were included in sections A and B; the most frequent dominant alleles were also considered.

## 5. Examples of Unusual Genotype–Phenotype Correlations for Slavic Pathogenic Alleles

In some cases, specific pathogenic alleles may cause unusual phenotypic features that alter the clinical manifestation of the disease. For example, Slavic mutation c.1642C>T (p.Gln548*) in the *BLM* gene, which has been detected in 0.2–0.6% residents of Slavic countries, is associated with an atypical presentation of Bloom syndrome, i.e., the absence of sun-induced erythema [207,208,209]. The unusually late onset of *NPHS2* Val290Met-associated steroid-resistant nephrotic syndrome probably leads to the underdiagnosis of this condition [128,130]. The Slavic variant in the *COL4A5* gene, c.1871G>A (p.Gly624Asp), is linked with a mild course of Alport syndrome [127]. The *BEST2* c.313G>C (p.Arg105Gly) variant, which is detected in Slovenian patients with Best vitelliform dystrophy type 2, is associated with variable retinal changes; some carriers of this allele have additional extramacular lesions [222]. 

The Ukrainian study of patients with dominant atypical corneal dystrophies revealed that individuals with *TGFBI* c.1673T>C (p.Leu558Pro) allele share the same haplotype and have an unusual corneal phenotype, combining granular and lattice lesions [223]. Interestingly, *TGFBI* c.1673T>C (p.Leu558Pro) substitution is also recurrently found in Spain; it has been proposed to classify the associated phenotype as a form of lattice corneal dystrophy (LCD), namely, LCD type IV [224]. The analysis of a large Spanish cohort has helped to confirm and extend the initial findings of Ukrainian investigators, who described the natural history of this disease. LCD type IV usually manifests with granular dot-like semitransparent opacities in the central cornea. Subsequently, along with the increase and eventual confluence of the granular lesions, the emergence of lattice lines is observed [223,224].

## 6. Contribution of Slavic Mutations to the Genetic Burden of Other Founder Communities: Ashkenazi Jews, Hutterites, and Mennonites

Ashkenazi Jews established settlements in Poland, Belarus, Ukraine, and other Slavic countries about 500 years ago; however, until the XX century, they continued to live mainly in relatively isolated communities. The life of Jewish people in Eastern Europe was dramatically affected by anti-Semitic sentiments; these hardships culminated in the Holocaust in which over 6 million Jewish people were exterminated by Nazis. There were several waves of mass migration of Ashkenazi Jews from Europe to North America in the XIX and XX centuries. After World War II, many Jewish people moved to Israel, which was established as an independent state in the year 1948. Despite Eastern European Jews were separated from Slavic people by religious, cultural, and legislative barriers, the genetic studies indicate a significant allelic exchange.

Some well-known Ashkenazi founder-mutations, such as *BRCA2* c.6174delT or *MSH2* c.1906G>C, are also recurrent in Slavic populations. One may suggest that their carriers have Jewish roots, however, more detailed investigations are required to examine this hypothesis. At the same time, many alleles, which were initially discovered in Ashkenazi people and, therefore, are often considered as Jewish genetic variants, appear to have a Slavic origin. The most known examples are *BRCA1* c.5382insC and *ATP7B* c.3207C>A (p.His1069Gln) mutations [225]. For some alleles (e.g., *BRCA1* c.5382insC, *BRCA2* c.6174delT, *MSH2* c.1906G>C), the origin and routes of dispersal have been already tracked by haplotyping of unrelated carriers and investigation of the haplotype structure [69,226,227]. In the absence of comprehensive haplotyping information, the spread of a given genetic variant can be tentatively traced by the analysis of its frequency gradient [40]. According to GnomAD database, Ashkenazi Jews are also characterized by high frequency of presumably Slavic pathogenic variants *C2* c.841_849 + 19del and *AP4B1* c.1160_1161del. North-European *CHEK2* 1100delC allele has similar frequencies in both Slavs (0.5%) [228] and Ashkenazi Jews (0.3%) [229]. Two major Slavic variants, *PAH* c.1222C>T (p.Arg408Trp) and *DHCR7* c.452G>A (p.Trp151*), are also recurrent in Ashkenazi Jewish people, although their frequency is lower when compared to the “true” Ashkenazi founder variants in this population [230,231,232]. There are examples of alleles showing regional founder effect in Slavs and being repeatedly identified in some Jewish patients. For example, “Serbian” *PAH* c.143T>C (p.Leu48Ser) mutation has been described in Tunisian, Caucasian and Bukharan Jewish patients [221,230]. It is essential to keep in mind that the information on the allele frequencies is not sufficient for the identification of the origin of a given variant, therefore, more precise genomic studies are warranted to validate the above speculations.

Some presumably Slavic alleles appear to be shared with other founder communities, possibly reflecting the geographical neighborhood at some periods of history. The *PCDH15* c.1103delT (p.Leu368Trpfs*58) variant is associated with Usher syndrome. It is known as a common cause of deaf-blindness in the Hutterites, a religious group that originated from the Swiss Anabaptist movement [233]. Fleeing from religious persecution, Hutterites migrated through the territory of the Czech Republic, Hungary, Slovenia, various Balkan states, Ukraine/Russian Empire, ending their three-hundred-year journey by emigration to North America at the end of the XIX century. In addition to Hutterites, the *PCDH15* c.1103delT (p.Leu368Trpfs*58) variant was detected only in Slovenian patients with Usher syndrome suggesting a Slavic origin of this allele in Hutterites [151]. The emergence of another community of Anabaptists, the Mennonites, is also related to several waves of migration; most of the Mennonites went further through the countries of Eastern Europe to Russia and beyond its borders to Central Asia. A small part of the descendants of Mennonites remains as residents of Russia and Kazakhstan. The *ATM* c.5932G>T (p.Glu1978*) allele, being the most common in Russia and Poland, is detected at high frequency in patients with ataxia–telangiectasia of Mennonite origin [203,204].

## 7. Exome- and Genome-Based Analysis of Genetic Load in Countries with Predominantly Slavic Population

Recurrent recessive pathogenic alleles may significantly influence the pattern of genetic diseases observed in a given population. While most of the recurrent variants have been historically detected via analysis of patients affected by a particular disease, recent ethnicity-based or country-based exome sequencing studies of healthy people have added a significant amount of relevant information. Apart from a number of small-scale efforts [54,156,234,235,236], there are several large genomic/exomic projects involving people of (Balto)-Slavic ancestry. Importantly, the GnomAD v.2.1.1 NFE cohort contains the subgroups of Bulgarians (n = 1335) and Estonians (n = 2418). 

A report describing the Thousand Polish Genomes database (n = 1222 genomes) has recently been released [195]. In addition to known Slavic/Polish founder variants in *TGM5*, *NBN*, *PROP1*, *NUP93*, *C19orf12*, etc., genes, this Polish project revealed the overrepresentation of a founder deletion in the *C2* gene (MAF = 0.025). A high frequency of structural variations involving the *MTMR2* gene, which are potentially associated with Charcot–Marie–Tooth 4B1 disease, has been observed [195].

The Russian exome database (n = 6096) suggests the unexpectedly high occurrence of *NEB* c.23989C>T (p.Arg7997Ter) (nemaline myodystrophy), *F7* c.995C>T (p.Ala332Val) (hemophilia), *OTOG* c.2464C>T (p.Gln822Ter) (autosomal-recessive hearing loss), *LIPA* c.894G>A (p.Gln298=) (cholesteryl ester storage disease), *SLC26A2* c.1957T>A (p.Cys653Ser) (autosomal recessive multiple epiphyseal dysplasia), *BCKDHB* c.832G>A (p.Gly278Ser) (maple syrup urine disease), and several other disease-associated variants [237]. These alleles demonstrated MAFs of 0.003–0.0068 in the Ruseq database compared to MAFs of 0.0002–0.0013 in the gnomAD NFE subgroup. Interestingly, *WDR35* c.1889T>G (p.Leu630Ter) allele, which is overrepresented in Russia (MAF = 0.0025, Ruseq database) and Estonia (MAF = 0.0029, gnomAD), has recently been reported as one of the two recurrent alleles in Polish patients with cranioectodermal dysplasia (Sensenbrenner syndrome) (25% of pathogenic alleles) [238].

In addition to investigations involving countries with predominantly Slavic populations, it is necessary to acknowledge a recent Estonian study. Expectedly, this large-scale exome sequencing effort (n = 2327) has identified many alleles, which are shared between Baltic and Slavic countries [239].

## 8. Recurrent Alleles in Slavic and Non-Slavic Populations

In most of instances, the association of a high frequency of particular pathogenic variants with specific ancestry is attributed to the role of population founders. However, the founder effect is usually observed in relatively small communities, which are biologically separated from their neighbors by geographic or cultural barriers. Countries with predominantly Slavic folks host more than two hundred million people, are spread across a huge territory, and do not have more barriers than other European communities; therefore, they do not share some essential characteristics typically observed in founder populations. However, several lines of evidence suggest that Slavic-speaking countries, particularly Eastern and some Western Slavs, demonstrate significantly higher levels of genetic homogeneity than many of their neighbors [12,240,241,242]. 

Some studies revealed that the genetic composition of European populations is associated mainly with their geographic location, while the linguistic or political borders play a secondary role in the gradients of allele frequencies [241,242,243,244]. However, there are some noticeable exceptions. For example, Poles demonstrate much more genetic similarity to other Eastern and Western Slavs compared with non-Slavic neighboring ethnicities (Germans, Scandinavians, Baltic peoples, Finns, etc.) in terms of genetic distance metrics [12,241,242]. Eastern Slavs (Ukrainians, Belarusians, and Russians) are generally distinct from closely located Finno-Ugric, Baltic, Uralic, and other non-Slavic ethnic groups, with the exception of Northern Russians [12,71]. The impact of geographic neighborhoods is more applicable to Czechs and Slovaks, who share genetic features of both Poles and Germans [12,242]. A similar trend is observed in Southern Slavs: while Croats and Slovenians (and to a lesser extent, Serbians) are more similar to their Western Slavic neighbors (Czechs and Slovaks), some other Southern Slavic-speaking communities are very close to non-Slavic-speaking Balkan populations (Romanians, Greeks, Hungarians) [12,242].

Our analysis of the scientific literature led to the identification of 150 recurrent pathogenic alleles, which are considerably more common in Slavs or certain Slavic ethnic groups than in other ethnicities. Some of these variants definitely, or most probably, have a pan-Slavic or even Balto-Slavic origin (n = 54), while others are of regional significance (Appendix A). The 52 alleles can be viewed as major Slavic alleles; i.e., they are a) relatively ancestry-specific, b) observed in two or more major subgroups of Slavs (Eastern, Western, and/or Southern Slavs), and c) contribute to a major (with arbitrarily chosen threshold of >25%) share of pathogenic alleles, identified in a corresponding patient cohort in one or several “Slavic” countries. The persistence of a major founder effect throughout a vast territory is unusual for large European countries. There are some pan-European major founder alleles, but their distribution is clinal and not confined to any ancestry; on the contrary, multiple regional-specific founder variants are known, which are observed in geographically small areas, probably reflecting the complex political history of Europe [245,246,247,248]. 

We attempted to compare the ratio between the number of known major recurrent pathogenic alleles and the size of the population for Slavic countries and well-known founder communities (Ashkenazi Jews, Finns, Icelanders, French Canadians; Appendix A). Ashkenazi Jews have approximately 119 recurrent disease-associated variants in a population consisting of 10–14 million people, so this ratio is within the range of 8.5 × 10^−6^–1.19 × 10^−5^. Similar scores are observed in Finns (7.1–8.3 × 10^−6^, 50 alleles per 6–7 million) and French Canadians (4.0–6.0 × 10^−6^, 40 alleles per 6.8–10 million). This estimate is an order of magnitude higher in Icelanders (9.0 × 10^−5^, 27 alleles per 0.3 million). Not surprisingly, Slavic people, being a larger population, demonstrate lower persistence of major founder pathogenic variants (1.9 × 10^−7^, 52 major founder alleles per 270 million; Appendix A). Obviously, the comparisons described above have significant limitations because some populations have been studied more intensively than others. For example, there is a systematic catalog of the so-called Finnish disease heritage, thanks to a specific nationwide research program carried out for several decades [249]. Ashkenazi Jewish genetics is even more extensively studied, and there is a specialized catalog of ancestry-specific variants in Israel [3]. Icelanders are extremely well investigated, including several exomic and genomic studies carried out in the recent past [250]. Rare multisystemic disorders presenting a diagnostic conundrum are generally better described in populations that have advanced genetic services and a research-oriented community of clinical geneticists (Appendix A).

## 9. Conclusions and Perspectives

The knowledge of recurrent alleles may facilitate the detection of genetic diseases in Slavic patients. For example, in many instances, the diagnosis of a hereditary condition can be established by the use of relatively cheap allele-specific PCR testing. In addition, PCR-based techniques can be utilized for the screening of some genetic disorders. The list of recurrent variants is likely to enlarge in the near future due to the increasing utilization of exome sequencing. Being relatively genetically homogeneous, Slavic populations provide opportunities for highly efficient genetic research and the identification of novel hereditary diseases. 

## Figures and Tables

**Figure 1 ijms-25-09335-f001:**
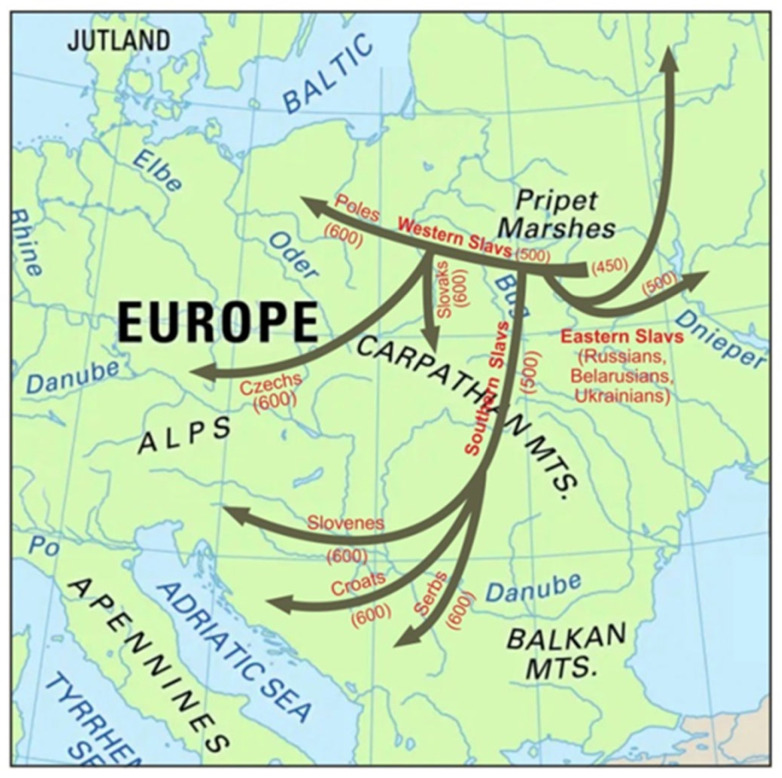
Historical dispersal of Slavic people. Approximate time of migration (years CE) is given in brackets.

**Figure 2 ijms-25-09335-f002:**
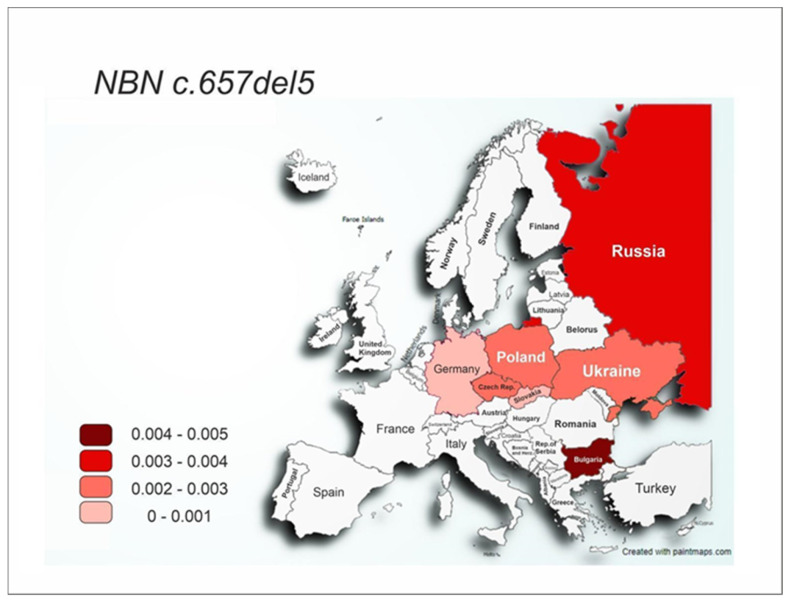
Minor allele frequency (MAF) of the *NBN* c.657del5 allele in Slavic and some non-Slavic countries (created with paintmaps.com, accessed on date 27 May 2024).

## Data Availability

No new data were created or analyzed in this study. Data sharing is not applicable to this article.

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
