# Peer review of "The Spectrum of Disease-Associated Alleles in Countries with a Predominantly Slavic Population"

_ijms, 2024, doi:10.3390/ijms25179335_

Round 1

Reviewer 1 Report

Comments and Suggestions for Authors

This review manuscript presents a comprehensive overview of the genetic landscape of Slavic populations, with a particular focus on the founder and recurrent pathogenic variants associated with hereditary diseases. The study highlights the unique opportunities for genetic research in these populations, stemming from the high degree of genetic homogeneity observed in many Slavic-speaking countries.

On the flipside:

The introduction provides a thorough background on the concept of founder populations, with clear references to well-known examples such as the Mennonites and Ashkenazi Jews. The manuscript successfully outlines the unique genetic attributes of Slavic populations, framing the discussion within the historical and demographic context of these communities. The methodology section effectively outlines the comprehensive literature review approach, including the PubMed search queries used to identify relevant studies. The inclusion of specific search terms and a detailed explanation of the criteria for including founder and recurrent variants enhances the rigor and transparency of the review. The results are presented clearly, with well-organized subsections covering various disease categories, such as inborn errors of metabolism, hereditary cancer syndromes, neurological diseases, and others. The manuscript includes detailed descriptions of specific founder mutations and recurrent alleles across different Slavic populations, providing valuable insights into the genetic epidemiology of these regions.

My recommendations:

The manuscript raises some intriguing hypotheses on the founder effect on mutations found in founder communities, such as the Ashkenazi Jews, the Hutterites and the Mennonites. However, since it is a report only of the mutations with limited haplotypic info, it is not a formal investigation. It resorts to a chicken-and-egg type of argument, and thus I would recommend to tone down the language on segment 6 (lines 576-624) with more mild expressions. For example the whole segment "are also recurrent in Slavic populations apparently reflecting Jewish roots in their carriers. At the same time, many alleles, which were initially discovered in Ashkenazi people and, therefore, are often considered as Jewish genetic variants, appear to have a Slavic origin" is way too strong in my opinion and it's not backed by any investigation on the author's part (eg a whole-genome analysis etc). It's very iteresting as a though, just make it more apparent that it is speculation and food for though, instead of a conclusion.

Just some instances of the correct nomenclature: Table 1. Just fix the name (Northern) Macedonia -> North Macedonia, as is the de jure name. Line 249: Macedonia -> North Macedonia. Line 290: Macedonia -> North Macedonia.

Overall, I liked it, it was a detailed and thorough manuscript and I think it is a great review of the genetics of rare variants in Slavic populations. Just these amendments and it will be good to go.

Author Response

Comment: For example the whole segment "are also recurrent in Slavic populations apparently reflecting Jewish roots in their carriers. At the same time, many alleles, which were initially discovered in Ashkenazi people and, therefore, are often considered as Jewish genetic variants, appear to have a Slavic origin" is way too strong in my opinion and it's not backed by any investigation on the author's part (e.g., a whole-genome analysis, etc.). It's very interesting as a though, just make it more apparent that it is speculation and food for though, instead of a conclusion.

Response: We have changed the phrasing in order to provide this information in a more balanced way: Some well-known Ashkenazi founder-mutations, such as BRCA2 c.6174delT or MSH2 c.1906G>C, are also recurrent in Slavic populations. One may suggest that their carriers have Jewish roots, however, more detailed investigations are required to examine this hypothesis. At the same time, many alleles, which were initially discovered in Ashkenazi people and, therefore, are often considered as Jewish genetic variants, appear to have a Slavic origin. The most known examples are BRCA1 c.5382insC and ATP7B c.3207C>A (p.His1069Gln) mutations [225]. For some alleles (e.g., BRCA1 c.5382insC, BRCA2 c.6174delT, MSH2 c.1906G>C), the origin and routes of dispersal have been already tracked by haplotyping of unrelated carriers and investigation of the haplotype structure [69,226,227]. In the absence of comprehensive haplotyping information, the spread of a given genetic variant can be tentatively traced by the analysis of its frequency gradient [40]. According to GnomAD database, Ashkenazi Jews are also characterized by high frequency of presumably Slavic pathogenic variants C2 c.841_849 + 19del and AP4B1 c.1160_1161del. North-European CHEK2 1100delC allele has similar frequencies in both Slavs (0.5%) [228] and Ashkenazi Jews (0.3%) [229]. Two major Slavic variants, PAH c.1222C>T (p.Arg408Trp) and DHCR7 c.452G>A (p.Trp151*), are also recurrent in Ashkenazi Jewish people, although their frequency is lower when compared to the “true” Ashkenazi founder variants in this population [230–232]. There are examples of alleles showing regional founder effect in Slavs and being repeatedly identified in some Jewish patients. For example, “Serbian” PAH c.143T>C (p.Leu48Ser) mutation has been described in Tunisian, Caucasian and Bukharan Jewish patients [230]. It is essential to keep in mind that the information on the allele frequencies is not sufficient for the identification of the origin of a given variant, therefore, more precise genomic studies are warranted to validate the above speculations”.  

Comment: Just some instances of the correct nomenclature: Table 1. Just fix the name (Northern) Macedonia -> North Macedonia, as is the de jure name. Line 249: Macedonia -> North Macedonia. Line 290: Macedonia -> North Macedonia.

Response: Done.

Reviewer 2 Report

Comments and Suggestions for Authors

The research of Slavic genetic diseases is less extensive than that of Western Europe and North America, highlighting the need for more exploration and knowledge of the genetic landscape in these groups. These groups, similar to a huge founding community, provide important possibilities for genetic study. The study identifies important pathogenic variants frequent among Slavs, including ATP7B c.3207C>A and PAH c.1222C>T, as well as 52 pan-Slavic germ-line mutations such as NBN c.657_661del and BRCA1 c.5266dupC. Specific Slavic communities have their own unique mutations, such as the Polish SDHD c.33C>A and the Russian ARSB c.1562G>A. However, the article contains several inconsistencies.

The article includes the phrase "countries with predominantly Slavic populations" while listing "Russia, Ukraine, Poland, Belarus, Czech Republic, Serbia, Bulgaria, Slovakia, Croatia, Bosnia & Herzegovina, Slovenia, Northern Macedonia, Montenegro" (lines 69 through 71). However, it wrongly refers to North Macedonia as "Northern Macedonia" and includes Bosnia and Herzegovina, which has a large non-Slavic population.

The article states that "the first mention of the Slavs appeared in Byzantine chronicles and is dated by the VI century AD" (line 58). Later, it says, "the division for Eastern Slavs, Western Slavs... occurred around the V-VI centuries AD" (line 62). Both of these assertions should use AD or CE consistently.

The article indicates that "the Eastern Slavs were divided into Russians, Ukrainians, and Belarusians at the end of the Middle Ages" (line 66), however this partition might have occurred much earlier, rather than at the end of the Middle Ages.

Several census data citations are either out of date or do not correspond to the correct years. For example, the Polish census data from 2011 (line 82) should be verified for accuracy. Furthermore, the Ukrainian census data is from 2001 (line 101), which is rather old. Also, the table of Slavic populations (lines 105-111) has some out-of-date references and should be updated with the most recent census data. The population percentages do not always match the total population statistics. For example, Belarus's 2019 statistics shows 9,145,526 Slavs out of 9,413,446 total population, representing 97.2%. However, the precise population counts should be double-checked for accuracy.

The article refers to the BRCA1 c.5266dupC (p.Gln1756ProfsX74) variant as the "most common BRCA1/2 alteration in the world" (line 243). This statement needs clarification as it is specific to certain populations, particularly those of Eastern European descent, and not necessarily the entire world.

According to the article, the hypomorphic mutation c.1871G>A (p. Gly624Asp) in the COL4A5 gene affects 35% of patients in Slovenia, 33% in Cyprus, and 30% in Hungary, but is less common in Russia (16%) and Australia (11%). Given that Alport syndrome is an uncommon condition, these high rates appear improbable without more epidemiological context or evidence.

According to the paper, the NPHS2 c.868G>A (p.Val290Met) mutation "usually develops the disease in adult age" whereas conventional NPHS2-associated SRNS appears in childhood. This difference is unclear, and it calls into doubt the variant's classification as "hypomorphic."The origin of the NUP93 c.1772G>T (p.Gly591Val) mutation is mentioned as undetermined despite stating that all carriers share the same haplotype. If haplotype information is available, it might provide clues about the origin.

The prevalence rates for numerous mutations (e.g., SLC7A9 c.313G>A) are reported with large ranges (11-50%), indicating either a very different dataset or inadequate data for precision calculation. The article does not explain the difference. In addition, the frequency ranges provided for the LDLR c.1775G>A (p.Gly592Glu) variation (9-22%) in different groups should be accompanied with a standard deviation or confidence interval to demonstrate the variability and dependability of these values.

The assertion that the GJB2 c.35delG variant is the primary genetic cause of hearing loss in all European populations should be supported by more current and thorough investigations.

The statement "relatively high frequency of genetic alterations" (line 193) is ambiguous, and "relatively high" should be defined quantitatively.

The phrase "many alleles, which were initially discovered in Ashkenazi people and, therefore, are often considered as Jewish genetic variants, appear to have a Slavic origin" requires more solid genetic evidence and references to particular research demonstrating this allele interchange. The percentages stated for the incidence of the hypomorphic mutation in different groups appear to be rather high and may require further testing or context to make epidemiological sense.

Minor revision

The methodology section references the inclusion of publications published before January 5, 2024 (line 120), which is a future date that should be changed.

The article utilises several words for the same ethnic groups, such as "Byelorussian" and "Belarusian" (lines 132-135). The terminology should be consistent throughout the article.

It is required to replace the word mutation with pathogenic variant.

There are various examples of repeated information. For example, the line 397 "The phenotype of this disease is intermediate between so-called basement membrane disease (BMTD), which is a benign condition, and severe Alport syndrome" is reminiscent of previous comments of the disease severity scale.

Author Response

Comment: The article includes the phrase "countries with predominantly Slavic populations" while listing "Russia, Ukraine, Poland, Belarus, Czech Republic, Serbia, Bulgaria, Slovakia, Croatia, Bosnia & Herzegovina, Slovenia, Northern Macedonia, Montenegro" (lines 69 through 71). However, it wrongly refers to North Macedonia as "Northern Macedonia" and includes Bosnia and Herzegovina, which has a large non-Slavic population.

Response: We have corrected the mistake in name of the state of North Macedonia. However, Bosnia and Herzegovina has a predominantly Slavic population. The largest ethnic group in Bosnia and Herzegovina are Bosniaks, which have a Southern Slavic origin, however, contrary to the majority of Slavic people, they largely belong to the Muslim religious community [for example, see Friedman F. The Muslim Slavs of Bosnia and Herzegovina (with Reference to the Sandžak of Novi Pazar): Islam as National Identity. Nationalities Papers. 2000;28(1):165-180. doi:10.1080/00905990050002498]. In addition, Bosnia and Herzegovina hosts a large number of Serbs and Croats.

Comment: The article states that "the first mention of the Slavs appeared in Byzantine chronicles and is dated by the VI century AD" (line 58). Later, it says, "the division for Eastern Slavs, Western Slavs... occurred around the V-VI centuries AD" (line 62). Both of these assertions should use AD or CE consistently.

Response: We now use CE in all instances.

Comment: The article indicates that "the Eastern Slavs were divided into Russians, Ukrainians, and Belarusians at the end of the Middle Ages" (line 66), however this partition might have occurred much earlier, rather than at the end of the Middle Ages.

Response: We have corrected this statement: “During the Middle Ages the Eastern Slavs divided into Russians, Ukrainians and Belarusians”.

Comment: Several census data citations are either out of date or do not correspond to the correct years. For example, the Polish census data from 2011 (line 82) should be verified for accuracy. Furthermore, the Ukrainian census data is from 2001 (line 101), which is rather old. Also, the table of Slavic populations (lines 105-111) has some out-of-date references and should be updated with the most recent census data. The population percentages do not always match the total population statistics. For example, Belarus's 2019 statistics shows 9,145,526 Slavs out of 9,413,446 total population, representing 97.2%. However, the precise population counts should be double-checked for accuracy.

Response: We have updated the census data for Poland, Czech Republic, Serbia, Bulgaria, Slovakia, Croatia and North Macedonia. We have also checked the census for Ukraine and ensured that the data from 2001 is the most recent resource available. We have double-checked Belarus`s 2019 statistics:  when we sum 7990719 Belarusians, 706992 Russians, 287693 Poles, 159656 Ukrainians and 466 Bulgarians, we get 9,145,526 Slavs out of 9,413,446 total population.  

Comment: The article refers to the BRCA1 c.5266dupC (p.Gln1756ProfsX74) variant as the "most common BRCA1/2 alteration in the world" (line 243). This statement needs clarification as it is specific to certain populations, particularly those of Eastern European descent, and not necessarily the entire world.

Response: We have changed this statement to “…is particularly common in populations of Eastern European descent”.

Comment: According to the article, the hypomorphic mutation c.1871G>A (p. Gly624Asp) in the COL4A5 gene affects 35% of patients in Slovenia, 33% in Cyprus, and 30% in Hungary, but is less common in Russia (16%) and Australia (11%). Given that Alport syndrome is an uncommon condition, these high rates appear improbable without more epidemiological context or evidence.

Response: Apparently, some misunderstanding has occurred with regard to this issue. We have modified the wording to avoid ambiguity: “Hypomorphic mutation c.1871G>A (p. Gly624Asp) in the COL4A5 gene is detected in a high proportion of patients with X-linked dominant Alport syndrome (XLD-AS) residing in Southern and Central Europe. Most often this mutation occurs among XLD-AS patients from Slovenia (35%), Cyprus (33%), Hungary (30%); it is also found in patients from Russia (16%) and Australia (11%) [121–126].

Comment: According to the paper, the NPHS2 c.868G>A (p.Val290Met) mutation "usually develops the disease in adult age" whereas conventional NPHS2-associated SRNS appears in childhood. This difference is unclear, and it calls into doubt the variant's classification as "hypomorphic.".

Response: Regarding NPHS2 c.868G>A (p.Val290Met), we refer to papers [Thomasová et al., The most common founder pathogenic variant c.868G > A (p.Val290Met) in the NPHS2 gene in a representative adult Czech cohort with focal segmental glomerulosclerosis is associated with a milder disease and its underdiagnosis in childhood. Front. Med. (Lausanne). 2023, 10, 1320054, doi: 10.3389/fmed.2023.1320054 and Kerti et al. NPHS2 p.V290M mutation in late-onset steroid-resistant nephrotic syndrome. Pediatr. Nephrol. 2013, 28, 751–757, 10.1007/s00467-012-2379-2], which put an emphasis on the adult onset of NPHS2 p.V290M-associated SRNS.

Comment:  The origin of the NUP93 c.1772G>T (p.Gly591Val) mutation is mentioned as undetermined despite stating that all carriers share the same haplotype. If haplotype information is available, it might provide clues about the origin.

Response: We removed the statement on undetermined origin of the NUP93 c.1772G>T (p.Gly591Val) mutation.

Comment: The prevalence rates for numerous mutations (e.g., SLC7A9 c.313G>A) are reported with large ranges (11-50%), indicating either a very different dataset or inadequate data for precision calculation. The article does not explain the difference. In addition, the frequency ranges provided for the LDLR c.1775G>A (p.Gly592Glu) variation (9-22%) in different groups should be accompanied with a standard deviation or confidence interval to demonstrate the variability and dependability of these values.

Response: These ranges reflect actual published estimates. We have added a comment on this issue: “It is necessary to acknowledge, that many of these variants demonstrate huge interstudy variations with regard to their frequency. Differences in ethnic or geographical origin of the patients, small study size, selection bias, or technical limitations may substantially contribute to these inconsistencies, however, the analysis of involved confounding factors is beyond the scope of this review.”

Comment: The assertion that the GJB2 c.35delG variant is the primary genetic cause of hearing loss in all European populations should be supported by more current and thorough investigations.

Response:  We have deleted the word “all” to soften this statement. In addition, we have added a reference to a comprehensive study describing the distribution of this allele [Chan DK, Chang KW. GJB2-associated hearing loss: systematic review of worldwide prevalence, genotype, and auditory phenotype. Laryngoscope. 2014 Feb;124(2):E34-53. doi: 10.1002/lary.24332].

Comment: The statement "relatively high frequency of genetic alterations" (line 193) is ambiguous, and "relatively high" should be defined quantitatively.

Response: We have modified this statement: “Slavic countries are characterized by a number of recurrent genetic alterations, which affect genes involved in the assembly of the mitochondrial respiratory chain components”

Comment: The phrase "many alleles, which were initially discovered in Ashkenazi people and, therefore, are often considered as Jewish genetic variants, appear to have a Slavic origin" requires more solid genetic evidence and references to particular research demonstrating this allele interchange.

Response: We believe, we do support this statement by appropriate comments and references: “The most known examples are BRCA1 c.5382insC and ATP7B c.3207C>A (p.His1069Gln) mutations [225]. The origin and routes of allele dispersal can be revealed by haplotyping of unrelated carriers and investigation of the haplotype structure, as exemplified by the studies of BRCA1 c.5382insC, BRCA2 c.6174delT or MSH2 c.1906G>C mutations [69,226,227]. In the absence of comprehensive haplotyping information, the spread of a given genetic variant can be tentatively traced by the analysis of its frequency gradient [40]. According to GnomAD database, Ashkenazi Jews are also characterized by high frequency of presumably Slavic pathogenic variants C2 c.841_849 + 19del and AP4B1 c.1160_1161del. North-European CHEK2 1100delC allele has similar frequencies in both Slavs (0.5%) [228] and Ashkenazi Jews (0.3%) [229]. Two major Slavic variants, PAH c.1222C>T (p.Arg408Trp) and DHCR7 c.452G>A (p.Trp151*), are also recurrent in Ashkenazi Jewish people, although their frequency is lower when compared to the “true” Ashkenazi founder variants in this population [230–232]”.

Comment: The percentages stated for the incidence of the hypomorphic mutation in different groups appear to be rather high and may require further testing or context to make epidemiological sense.

Response: The incidence of hypomorphic mutation associated with X-linked dominant Alport syndrome (XLD AS) is provided as a share of this variant among all pathogenic alleles in patients with genetically confirmed XLD AS. More detailed analysis of the available data is beyond the scope of this review.   

Comment: The methodology section references the inclusion of publications published before January 5, 2024 (line 120), which is a future date that should be changed.

Response:  January 5, 2024 is the past date.

Comment: The article utilizes several words for the same ethnic groups, such as "Byelorussian" and "Belarusian" (lines 132-135). The terminology should be consistent throughout the article.

Response: Those words had been mentioned as search terms. We have to take into account alternative spellings utilized in research literature (i.e., Belarusian, Belorusian, Belorussian, Belarus, and Byelorussian), and consider them all while looking for relevant publications.

Comment: It is required to replace the word mutation with pathogenic variant.

Response: We recognize that the term “pathogenic variant” is more accurate according to ACMG guidelines. Nevertheless, “mutation” is still widely used in current literature, especially in certain contexts. For example, “founder mutation” is still a more commonly used term, than “founder pathogenic variant”. We now comment on this in the Methodology section.

Comment: There are various examples of repeated information. For example, the line 397 "The phenotype of this disease is intermediate between so-called basement membrane disease (BMTD), which is a benign condition, and severe Alport syndrome" is reminiscent of previous comments of the disease severity scale.

Response: There was no mention of Alport syndrome before the line 397. We briefly repeat the information on the mild course of the Alport syndrome in a section where we specifically discuss “Slavic” alleles associated with the atypical disease manifestation (Section 5).

Reviewer 3 Report

Comments and Suggestions for Authors

The manuscript examines genetic research conducted on Slavic populations, with a specific emphasis on disease-related genes and their frequency. The text emphasizes the historical and genetic separation that has led to a distinct genetic makeup in these communities, focusing specifically on the influence of Slavic alleles on disorders such as Bloom's syndrome and Alport syndrome. Furthermore, it investigates the genetic overlap between Slavic populations and other ethnic groups, such as Ashkenazi Jews and Hutterites, with a focus on the common and distinct genetic indicators.

Here are some critical scientific questions to consider for the manuscript:

1. How did the researchers control for potential biases or confounders in their review of genetic data from Slavic populations, especially given the varied historical migrations and intermixtures reported among these groups?
2. Could the authors provide further explanation regarding the techniques employed to ascertain the occurrence of the "founder effect" in distinct alleles throughout the Slavic populations? Specifically, how did they differentiate between alleles that were disseminated by migration and those that arose from independent mutations?
3. What statistical approaches were used to evaluate the significance and trustworthiness of the reported frequencies of disease-associated alleles among various Slavic groups?
4. In light of the intricate historical interplay between Slavic and non-Slavic groups, how did the authors take measures to safeguard the genetic characteristics associated with Slavic heritage from being altered by external genetic factors?
5. The publication discusses the utilization of haplotyping as a means to identify alleles of Slavic descent. Could the authors maybe elaborate on the haplotyping approaches employed and analyze their precision and constraints within this particular genetic framework?
6. Given the unique genetic traits linked to the Slavic populations, how do the authors differentiate between pathogenic and non-pathogenic variants in the context of clinical relevance?
7. Can the authors provide more details about the approaches employed to ensure the precision of haplotype mapping, specifically in differentiating between alleles that are common among various ethnic groups?
8. What are the possible consequences of the findings for clinical procedures in Slavic nations, particularly in relation to genetic counseling and methods for disease prevention?
9. How could the results of the study impact future genetic research or healthcare policies in Slavic and non-Slavic countries, taking into account the identified shared genetic characteristics and variations? 

Author Response

Comment: How did the researchers control for potential biases or confounders in their review of genetic data from Slavic populations, especially given the varied historical migrations and intermixtures reported among these groups?

Response: We comment on this issue: The origin and routes of allele dispersal can be revealed by haplotyping of unrelated carriers and investigation of the haplotype structure, as exemplified by the studies of BRCA1 c.5382insC, BRCA2 c.6174delT or MSH2 c.1906G>C mutations [69,226,227]. In the absence of comprehensive haplotyping information, the spread of a given genetic variant can be tentatively traced by the analysis of its frequency gradient [40].”

Comment: Could the authors provide further explanation regarding the techniques employed to ascertain the occurrence of the "founder effect" in distinct alleles throughout the Slavic populations? Specifically, how did they differentiate between alleles that were disseminated by migration and those that arose from independent mutations?

Response: We have added an appropriate comment to the Methodology section: The formal proof of founder nature of the variant requires haplotyping, which has not yet been performed for all variants mentioned in this article”.

Comment: What statistical approaches were used to evaluate the significance and trustworthiness of the reported frequencies of disease-associated alleles among various Slavic groups?

Response: We now acknowledge significant variations in the reported frequencies of recurrent pathogenic alleles: “It is necessary to acknowledge, that many of these variants demonstrate huge interstudy variations with regard to their frequency. Differences in ethnic or geographical origin of the patients, small study size, selection bias, or technical limitations may substantially contribute to these inconsistencies, however, the analysis of involved confounding factors is beyond the scope of this review”.

Comment: In light of the intricate historical interplay between Slavic and non-Slavic groups, how did the authors take measures to safeguard the genetic characteristics associated with Slavic heritage from being altered by external genetic factors?

Response: We believe that Table 2 addresses this question. It describes 3 categories of pathogenic variants [ A) Pan-European alleles demonstrating increased frequency in Slavic populations; B) “Pan-Slavic” alleles shared by at least two Slavic communities, but infrequently or never reported in most of other populations; C) “Regional” Slavic pathogenic alleles, infrequently or never reported in other populations ]. Obviously, the role of external genetic contribution cannot be excluded for each particular allele, however, the Slavic origin of the vast majority of described variants appears to be beyond reasonable doubt.

Comment: The publication discusses the utilization of haplotyping as a means to identify alleles of Slavic descent. Could the authors maybe elaborate on the haplotyping approaches employed and analyze their precision and constraints within this particular genetic framework?

Response: We systematically reference all studies, which involved haplotyping (see the text of the paper and Supplementary Table S1). The corresponding information regarding haplotyping approaches can be found in the original publications.   

Comment: Given the unique genetic traits linked to the Slavic populations, how do the authors differentiate between pathogenic and non-pathogenic variants in the context of clinical relevance?

Response: We applied ACMG criteria to the original data described in the referenced articles:  Apart from pathogenic (P) or likely pathogenic variants (LP), we considered 11 variants with conflicting interpretations and 17 variants of uncertain significance, which can be reclassified to P/LP categories according to the ACMG criteria“.

Comment: Can the authors provide more details about the approaches employed to ensure the precision of haplotype mapping, specifically in differentiating between alleles that are common among various ethnic groups?

Response:  We systematically reference all studies, which involved haplotyping (see the text of the paper and Supplementary Table S1). The corresponding information regarding haplotyping approaches can be found in the original publications.   

Comment: What are the possible consequences of the findings for clinical procedures in Slavic nations, particularly in relation to genetic counseling and methods for disease prevention?
Response: We now comment on this issue in the Conclusions and Perspectives section: “The knowledge on recurrent alleles may facilitate the detection of genetic diseases in Slavic patients. For example, in many instances the diagnosis of hereditary condition can be established by the use of a relatively cheap allele-specific PCR testing. In addition, PCR-based techniques can be utilized for the screening of some genetic disorders”.

Comment: How could the results of the study impact future genetic research or healthcare policies in Slavic and non-Slavic countries, taking into account the identified shared genetic characteristics and variations? 

Response: Please see the revised version of Conclusions and Perspectives section. 

Round 2

Reviewer 2 Report

Comments and Suggestions for Authors

All the necessary changes were performed by modifying the text and responding point-by-point to the discrepancies  found. In my opinion, there is no need for any more alterations.

Reviewer 3 Report

Comments and Suggestions for Authors

The authors have responded to my comments and questions satisfactorily.